# RNA Interference Strategies for Future Management of Plant Pathogenic Fungi: Prospects and Challenges

**DOI:** 10.3390/plants10040650

**Published:** 2021-03-29

**Authors:** Daniel Endale Gebremichael, Zeraye Mehari Haile, Francesca Negrini, Silvia Sabbadini, Luca Capriotti, Bruno Mezzetti, Elena Baraldi

**Affiliations:** 1Department of Agricultural and Food Sciences (DISTAL), University of Bologna, viale Fanin 44, 40126 Bologna, Italy; danend2000@yahoo.com (D.E.G.); zerayemehari.haile2@unibo.it (Z.M.H.); francesca.negrini6@unibo.it (F.N.); 2Ethiopian Institute of Agricultural Research (EIAR), P.O. Box 2003, Addis Ababa 1000, Ethiopia; 3Department of Agricultural, Food and Environmental Sciences, Marche Polytechnic University, 60131 Ancona, Italy; s.sabbadini@staff.univpm.it (S.S.); l.capriotti@pm.univpm.it (L.C.); b.mezzetti@staff.univpm.it (B.M.); 4Research Group on Food, Nutritional Biochemistry and Health, Universidad Europea del Atlántico, 39011 Santander, Spain

**Keywords:** RNA interference, dsRNA delivery, small RNA production, dsRNA formulation

## Abstract

Plant pathogenic fungi are the largest group of disease-causing agents on crop plants and represent a persistent and significant threat to agriculture worldwide. Conventional approaches based on the use of pesticides raise social concern for the impact on the environment and human health and alternative control methods are urgently needed. The rapid improvement and extensive implementation of RNA interference (RNAi) technology for various model and non-model organisms has provided the initial framework to adapt this post-transcriptional gene silencing technology for the management of fungal pathogens. Recent studies showed that the exogenous application of double-stranded RNA (dsRNA) molecules on plants targeting fungal growth and virulence-related genes provided disease attenuation of pathogens like *Botrytis cinerea*, *Sclerotinia sclerotiorum* and *Fusarium graminearum* in different hosts. Such results highlight that the exogenous RNAi holds great potential for RNAi-mediated plant pathogenic fungal disease control. Production of dsRNA can be possible by using either in-vitro or in-vivo synthesis. In this review, we describe exogenous RNAi involved in plant pathogenic fungi and discuss dsRNA production, formulation, and RNAi delivery methods. Potential challenges that are faced while developing a RNAi strategy for fungal pathogens, such as off-target and epigenetic effects, with their possible solutions are also discussed.

## 1. Introduction

Pathogens have decreased the productivity of crops since the advent of agriculture, and farmers have been exploring ways of safeguarding their crops from these organisms. The use of synthetic pesticides is currently an indispensable means of intensive agricultural systems to guarantee food supply worldwide, protecting crops from pathogens, which otherwise would cause more than 30% yield losses [1,2]. There is a long tradition of using synthetic pesticides which have been developed and applied to control pathogens. However, the evolution of pathogens resistance to pesticides, together with the concern for the environment and human health, has stimulated demand for more selective, environmentally friendly, and cost-effective alternative control methods for pathogens and pests [3]. Scientists have allocated a great deal of intellectual energy into seeking alternative strategies to reduce crop losses, such as the development of tolerant/resistant plants to pathogens and pests and with increased quality products by using conventional breeding and plant biotechnological tools [4]. More recently, gene silencing through RNA interference (RNAi) is offering a new opportunity for precision breeding and for the development of new products for protecting plants from pathogens and pests. RNAi is a conserved eukaryotic mechanism triggered by double-stranded RNA (dsRNA) molecules. It is associated with diverse eukaryotic regulatory processes, including protection against viral infection, control of transposon movement, regulation of genome stability, gene expression, and heterochromatin formation [5,6].

RNAi was first reported by Napoli and colleagues [7] to produce violet petunias, the *chalcone synthase* gene (*CHS*), encoding for a key enzyme in flavonoid biosynthesis, was overexpressed by introducing a transgene that resulted in an unintended white petunia phenotype. Further analysis revealed declined expression of both the endogenous and exogenously introduced *CHS* gene, which led to the conclusion that the transgene co-suppressed the endogenous *CHS* gene. A similar phenomenon was reported in the filamentous fungus *Neurospora crassa* [8], where the introduction of the transgene ‘*albino-1*’ resulted in the quelling of the endogenous gene. Similarly, in *Caenorhabditis elegans*, the injection of dsRNAs led to the silencing of *unc-22* gene, highly homologous in sequence to the delivered dsRNA molecules [9]. Over the last two decades, the understanding of RNAi has evolved from initial observation of unexpected patterns of expression to a deeper understanding of a multi-faceted network of mechanisms that regulate gene expression in many organisms [10,11,12]. Consequently, RNAi is getting research attention also as an environmentally friendly alternative to agricultural pest and pathogen control. In fact, because of its sequence-dependent mode of action, RNAi technology has an enormous range of potential as plant protection application, including control against insects [13], mite pests [14,15], plant pathogens [11,16,17,18], nematodes, and weeds [10,19,20,21]. 

The concept is based on the administration of small RNA (dsRNA/siRNA) molecules that induce the silencing of key genes in pathogenic organisms, thereby limiting/stopping their growth. Delivering dsRNAs to a target organism is a crucial aspect that determines the success of the RNAi technology in crop protection. Delivery can be achieved through host-induced gene silencing (HIGS) RNAi approach, corresponding to in-planta expression of siRNA targeting key genes of the pest/pathogen. Besides HIGS, exogenous delivery of dsRNA can be considered as an alternative approach. In this review selected research findings on RNAi approaches through exogenous delivery of small RNA molecules targeting plant pathogenic fungi will be discussed. Small RNA production techniques, potential limitations, and solutions for the application of RNAi for fungal disease control are also discussed.

## 2. RNAi for Resistance against Plant Pathogenic Fungi

In the past, RNAi in plants has been mainly used to improve resistance to diseases by silencing susceptibility genes, those genes that negatively regulate plant defense responses [22]. During the last decade, however, RNAi has been more exploited to provide plants with so-called “pathogen-derived resistance”, where resistance is achieved through small interfering RNAs (siRNAs) able to silence genes that are important for infection or the life cycle of the pathogen [23,24,25]. The silencing process starts with the cleavage of dsRNAs into 21–25-nucleotide-long double-stranded siRNAs in cytoplasm by Dicer or Dicer-like homologs and sRNA-specific RNase III family enzyme. Dicer protein contains an N-terminal helicase domain, a Piwi/Argonaute/Zwille (PAZ) motif, a dsRNA binding domain, and two RNase III motifs at the C-terminus. Dicer-generated siRNAs are then incorporated into a multi-component protein complex, the RNA-induced silencing complex (RISC), which becomes activated on ATP-dependent unwinding of the siRNA duplex [26]. RISC contains an Argonaute protein that has a sRNA-binding domain and an endo-nucleolytic activity for cleavage of target RNAs [26]. Once the siRNA is incorporated into RISC, it will be unzipped into the guide and passenger strands, the latter will be degraded, and the guide strand will bind to the target mRNA sequence and stimulate its endo-nucleolytic cleavage or will inhibit translation [27]. Although greatly diminished, residual mRNA levels can be detected. Therefore, the RNAi-mediated silencing of a particular gene is commonly referred to as a ‘knockdown’ rather than a ‘knockout’ [28,29]. Within the fungal kingdom, the mechanistic facets of RNAi were studied in *N. crassa* [8,30]. Since then, RNAi machinery has been recognized in a wide range of fungal species. The use of RNAi as a tool for reverse genetics, targeted at modification of fungal gene expression, is continually growing with a large number of fungal species already proved to be responsive [31]. Furthermore, the functionality of absorbed exogenous RNAi molecules offers excellent adaptability and flexibility in securing the required effects on gene expression of fungi, even without the need to genetically modify the targeted pathogen [11,32]. This homology-based gene silencing stimulated by transgenes (co-suppression), antisense, or dsRNAs has been demonstrated in several plant pathogenic fungi/oomycetes, including different mold fungi, such as *Botrytis cinerea*, *Neurospora crassa*, and *Sclerotinia sclerotiorum* [11,18,33,34]; blast, blight, and rust fungi, such as *Fusarium asiaticum*, *Fusarium graminearum*, *Magnaporthe oryzae,* and *Puccinia striiformis f. sp. tritici* [17,35,36,37,38]; mildew, and others, such as *Blumeria graminis*, *Cochliobolus sativus,* and *Venturia inaequalis* [39,40,41]. Over the past few years, a variety of target genes have been used to test whether RNAi is functional in plant-fungal pathogens (Table 1). To date, the number of successful candidate genes studied that led to reduced fungal growth development is limited, and includes effectors, cell wall elongation, chitinase, and hexose transporter genes. Much work remains to be done to identify suitable fungal candidate genes. Fortunately, opportunities exist to establish high-throughput screening pipelines to find strong candidates.

## 3. Small RNA Production Technologies

At present, exogenous application of dsRNA seems a new promising strategy to deploy RNAi for pathogen control in agriculture. To carry out exogenous approaches, silencing experiments have been successfully performed using sequence-specific small RNA molecules produced by different methods (Table 2). Production of dsRNAs can be possible by employing either in-vitro [11,17,99,100] or in-vivo synthesis [101,102]. Studies have shown that the application of in-vitro synthesized dsRNAs targeting essential fungal genes onto the plant leaf surface attenuated fungal infection by inhibiting fungal growth, altering fungal morphology, and reducing pathogenicity, leading to the development of weaker plant disease symptoms [11,17,33,37]. In-vitro methods consist of either enzymatic transcription or chemical synthesis with advantages and disadvantages for both. The enzymatic transcription approach is cost-effective for producing both short and long dsRNA molecules. This method is a source of pure dsRNA based on the annealing of two single-stranded (sense and antisense) RNAs (ssRNAs). Based on the principle of in-vitro transcription, on linearized DNA templates, or PCR-generated templates, the use of commercially available kits to produce dsRNA is widely used. Using in-vitro methods for dsRNA production, fungal resistance has been achieved in a plethora of cases as listed in Table 3. However, these kits are expensive when the production of large amounts of dsRNA is needed [17,103]. For RNAi studies on large-scale application, the enzymatic transcription method is therefore not a practical means of dsRNA production. Chemical synthesis, on the other hand, can produce a large yield of high purity dsRNA, but it is more expensive with the cost of synthesis increasing considerably as the length of the dsRNA increases [104]. Chemical synthesis of siRNA enables control over the quantity and purity of siRNA and it also allows chemical modifications to enhance stability, an important feature needed for delivery. Chemically synthesized siRNAs can be labeled for evaluating siRNA uptake or localization by fluorescence microscopy [105].

In-vivo production of dsRNA using genetically engineered bacteria (for ex. *Escherichia coli* and *Pseudomonas syringae*) and yeast (*Yarrowia lipolytica*) [110,111] emerged as an alternative approach to produce large quantities of dsRNAs at low cost. Concerning the costs, for example, it is possible to buy a fungus-derived dsRNA sequence produced in bacteria (*E. coli*) with about $1 USD per 1 g from low-cost companies [112]. These systems are able to produce large amounts of dsRNA molecules needed for field trial applications. Tenllado et al. [113] demonstrated that crude extracts of bacterially expressed dsRNAs are effective in protecting plants from virus infections when sprayed onto plant surfaces by a simple procedure. The use of recombinant bacteria to produce dsRNA is an efficient technique due to their ease of handling, ability to maintain plasmid, and the fast growth rate of bacteria [114]. Among the available *E. coli* strains, HT115 (DE3) is widely used to produce large amounts of dsRNA for exogenous application studies. The *E. coli* HT115 (DE3) harbors the pro-phage λDE3 encoding the Isopropyl β-D-1-thiogalactopyranoside (IPTG) inducible T7 polymerase gene for dsRNA transcription [101,113,115,116]. Even though, the bacterial production systems may contain bacterial homologous DNA molecules; that may affect the RNA quality and applicability, crude extracts of dsRNA can be applied on plants to test its efficiency against plant pathogens and pests [18,117]. Researchers demonstrated that bacterially expressed dsRNAs can be used to induce RNAi in fungus [18], virus [49], worms [118], and in insect pests [56,119]. Researchers are also using in-vivo dsRNA amplification employing *P. syringae* harboring the bacteriophage phi6 RNA-dependent RNA polymerase complex [120,121,122]. Niehl and colleagues [122] demonstrated that the in-vivo dsRNA production by *P. syringae* has great potential to allow therapeutic dsRNAs to be designed and produced for large-scale crop protection against different fungal and viral pathogens, and insect pests. However, the use of *E. coli* is still controversial because even if used as lysate containing the dsRNA, its residuals may have an impact on animal and human health [123]. Therefore, alternatives for expressing dsRNA in organisms are being explored, especially those that are generally considered safe for human consumption, which do not produce endotoxins or pose risks to health or the environment. One organism that possesses this characteristic is yeast (*Y. lipolytica*), which can provide unique advantages for the production of dsRNA. Alvarez-Sanchez et al. [111] observed that *Y. lipolytica* is a convenient host for producing and delivering dsRNA-ORF89 that can protect shrimp from white spot syndrome virus attack.

Besides other factors, the role of RNAi-based products for controlling fungal pathogens depends on the cost of production. Taking the cost trend into account, it is expected that small RNA production costs will decrease substantially in the future, with commercial companies investing in dsRNA production capacity. Over the past few years, a declining trend in the dsRNA production cost has been recorded. For example, the cost for producing 1 g of dsRNA using in-vitro nucleoside triphosphate (NTP) synthesis fell from $12,500 USD in 2008 to $60 USD in 2018 [16,124]. For field-scale pest and pathogen management, metric tons of dsRNA will be required. It is conceivable that such a huge demand cannot be satisfied only by an in-vitro dsRNA transcription system. For this reason, some industrial companies have achieved low-cost (almost $2 USD per 1 g of dsRNA) and large-scale production of dsRNA using bacteria [112,125]. 

## 4. Exogenous Delivery of Small RNA for Controlling Fungal Pathogens of Plants

The exogenous delivery method is certainly the most promising approach for the application of RNAi technology in the field [101,126]. This method avoids any modification of crop genomes and can be exploited against virtually any microbial pathogen that is responsive to RNAi approaches [11,127]. Hence, the exogenous method can be an alternative method to HIGS, more easily accepted by public and biosafety authority, and faster to optimize than the obtainment of a HIGS plant. The first observation, explaining exogenous delivery of dsRNA molecules on plants, inducing RNAi of a plant gene, was reported in *Nicotiana benthamiana* plants pre-treated with the surfactant Silwet L-77 [128]. In this study, in-vitro-transcribed 685 bp dsRNAs and/or chemically synthesized 21-nt sRNAs targeting the endogenous phytoene desaturase mRNA was sprayed on plant surfaces resulting in extensive phytoene desaturase downregulation [128]. In an exogenous RNAi mechanism, to induce RNAi and achieve successful protection against pathogens, two prerequisites are fundamental: i) the sensitivity of the target organism to the silencing process stimulated by dsRNA, and ii) the capability to uptake external RNA molecules from the environment by fungal pathogens [11,17,127], viruses [122,129,130], and insects [124,131,132]. Plants and fungi are capable of taking up externally applied dsRNAs and siRNAs. Reports showed that fungi can uptake 21nt sRNA duplexes as well as long dsRNAs of at least up to 800 nt [11,17]. The presence of Dicer, Argonaute, and RdRP proteins in several fungal species suggests that they should be capable to display active RNAi mechanisms [31,107,133]. However, exogenous delivery of small RNA to fungi can be tricky and for some fungal species has not been achieved yet. The reason underneath reluctance of RNA uptake by some fungal species can be difficult to explore and can be associated with different biological aspects, including the cell wall or membrane biochemical components [11]. For example, *Zymoseptoria tritici* encodes the core components of the RNAi machinery but still is dsRNA insensitive [23]. The authors have demonstrated through live-cell imaging that the conidiospores of *Z. tritici* were unable to absorb dsRNAs, suggesting that there may not be an encoded dsRNA receptor or a defect in the uptake pathway. Wang and co-workers reported rapid dsRNA uptake from the environment by *Botrytis cinerea* and that these RNAs were able to suppress fungal genes in a sequence-specific manner [11]. In *Sclerotinia sclerotiorum*, a scientific study demonstrated that the uptake of dsRNA occurs through clathrin-mediated endocytosis [134]. One of the few recent studies reported that various beneficial or pathogenic fungal and oomycetes organisms have diverse capacity to adsorb fluorescein-labeled dsRNA from the environment, and this competence seems to have an influence on the efficacy of the RNAi when virulence-related gene were targeted through a spray-induced gene silencing (SIGS) approach for the defense of the hosts. The authors showed that *Colletotrichum gloeosporioides* cannot uptake dsRNA, whereas in *Trichoderma virens* and *Phytophthora infestans* RNA uptake was limited. The situation is different in *Botrytis cinerea*, *Sclerotinia sclerotiorum*, *Rhizoctonia solani*, *Aspergillus niger*, and *Verticillium dahliae* in which fluorescent dsRNAs are already inside the fungal cells within 6 h after administration of specific long dsRNA [84]. Overall, information on dsRNA uptake in fungi is scarce, which is due to the limited number of studies conducted on the efficacy of exogenous RNAi against phytopathogenic fungi so far.

### 4.1. Formulation of Small RNA

The overall success of using exogenous RNAi is dependent on the mode of delivery of RNA molecules, application methods, length and/or concentration of dsRNAs, plant-organ specific activities, and stability under unsuitable environmental conditions [126,129,135,136]. The main constraint of exogenous applications of naked-dsRNAs is their short-term stability. Complexation of dsRNA with carrier molecules is a solution widely used to overcome this limitation [135,137,138]. Although most studies of dsRNA carriers for plant protection have concentrated on insects [139], the improved stability and penetrability of some formulations may also be applied to phytopathogenic fungi. It is tricky to predict when a fungal outbreak will occur and, thus, the longer the protective antifungal treatment on the surface of the plant will remain intact, the more likely it will be successful when the infection occurs. Furthermore, a variety of necrotrophic pathogens, such as *S. sclerotiorum*, can become systemic in a matter of days within the plant [140]. This underlines the importance of getting the optimized load of dsRNA into the fungus as quickly as possible, and this can be done by carriers that enhance penetrability. In order to increase stability and uptake efficiency, dsRNA can be incorporated into nanoparticles. Nanoparticles are the most common choice made in order to deliver the unstable naked dsRNA/siRNA to the targeted sites since they protect the dsRNA/siRNA from degradation. Besides, they can be used by adding target-specific ligands to their surface for targeted delivery [141]. Chitosan (poly β-1,4-Dglucosamine) is one of the most widely used polymers to generate nanoparticles to protect and deliver dsRNA/siRNA to target cells [142]. Chitosan has been the topic of many studies, due to its inexpensive production from marine waste, low toxicity, and a wide variety of molecular weights and modifications available [143,144]. It has been shown that chitosan-based formulations boost endonuclease stability and uptake in a variety of species of insects [145,146]. Another means to obtain an increased RNAi efficiency is through the use of layered double hydroxide clay nanosheets. Positively charged nanosheet stacks bind the dsRNA negative charges electrostatically and provide enhanced protection against environmental factors and nucleases. Mitter et al. [129] reported that loading RNAi inducing dsRNAs into layered double hydroxide clay nanosheets and applying to plant surface enabled sustained release of the dsRNA for up to 30 days. The formulated dsRNAs (Bioclay) offered protection against virus for up to 20 days post spraying, compared to naked dsRNA which offered 5 days protection window. Owing to this increased period of bioactivity, this technology also holds the potential to be useful in insect and fungal defense. Interestingly, this formulation also seems to facilitate uptake and systemic dissemination within the sprayed host plant [129]. The use of a class of very small nanoparticles, called carbon dots, for the delivery of siRNA to the *Nicotiana benthamiana* and tomato plants, has also been reported [147]. In addition, a liposome-based delivery method has been applied in insects, fungi, and nematodes [148,149,150] with success in altering gene expression and/or mortality. It should be stated here that, although carrier compounds considerably facilitate RNA delivery, they are also quite expensive and/or difficult to synthesize. Different administration strategies have been reported in mammalian cells, such as conjugation of dsRNAs to cholesterol, cationic lipids, and cell-penetrating peptides [151,152]. Future studies are required to determine whether they also improve dsRNA uptake and efficiency in fungal pathogens.

### 4.2. Delivery Methods

Different application/delivery strategies have been studied in various agricultural pest species and the main dsRNA application methods tested so far include high-pressure spray, injection into trunks, soil application, petiole absorption, brush-mediated application, infiltration, injection, root soaking, soil/root drench, and postharvest spraying of bunches [11,17,18,124,126,131,132,135,136,153,154]. When high-pressure spraying was used for the exogenous application of siRNAs, it was successful in inducing local and systemic silencing of the green fluorescent protein (GFP) transgene in *N. benthamiana* [126]. Here, high-pressure spraying was more effective compared to wiping, infiltration, and gene gun techniques. Direct exogenous application of dsRNA, by spreading with sterile individual soft brushes without using any additional techniques, was also observed successful in inducing efficient suppression of enhanced *green fluorescent protein* (*eGFP*) and *neomycin phosphotransferase–II* (*NPTII*) transgenes in *Arabidopsis* [136]. The authors analyzed the effects of different dsRNA concentrations (0.1, 0.35, and 1.0 μg/μl) and the concentration at 0.35 μg/μl had a higher significant influence on transgene-silencing efficiency [136]. The effects of different lengths of dsRNAs (315, 596, and 977-bp) targeting different virus genes were also investigated in *N. tabacum* leaves, and results indicated that shorter dsRNAs showed reduced antiviral activity, indicating that dsRNA length could influence its efficacy [155]. Overall, fungal uptake of environmental RNAs appears less dependent on RNA size, as both short sRNA duplexes and long dsRNAs are taken up and stimulate strong gene silencing in the fungal cells.

The efficient delivery of dsRNA is crucial in moving RNAi-based fungal control from laboratory to field. dsRNAs not only move within a fungus but they can also transfer from the environment to the fungus (environmental uptake), and between interaction of plants and fungus (cross-kingdom dsRNA trafficking), thereby subsequently inducing gene silencing in the fungal organism [134]. Exogenous RNAs derived from plant fungal pathogens gene sequences can either be directly internalized into fungal cells or indirectly via passage through plant tissue before transport into targeted fungal cells [11,17,106,156]. The vascular system of plants translocates RNAs [157]; indeed, RNAi in plants is linked with the production of a mobile signal that can move from cell-to-cell and over long distances. This fact can therefore be useful in the establishment of targeted strategies for the control of pathogens [158,159]. With respect to HIGS-in planta stable resistance, exogenous dsRNA applications offer shorter-term protection from fungal infections, but they could be particularly beneficial to shield agricultural food products during post-harvest storage and protecting plant species for which not defined nor efficient transformation protocols are available [127].

Studies conducted on exogenous RNAi concerning fungal pathogens, summarized in Table 3, showed that exogenous application is effective in suppressing fungal growth. For example, a recent study by Werner and colleagues [38] showed that using spray-induced gene silencing (SIGS), targeting *Argonaute* and *Dicer* genes of *F. graminearum*, afforded protection of barley leaves from infection by *F. graminearum*. Similarly, *F. asiaticum* virulence decreased when in-vitro-transcribed dsRNA targeting its *myosin 5* gene was sprayed on wounded wheat coleoptiles [106]. In another study, foliar applications of in-vitro transcribed dsRNAs on canola (*Brassica napus*), targeting 59 genes of necrotrophic fungi reduced *S. sclerotiorum* and *B. cinerea* leaves infection [34]. Spraying of detached barley leaves with dsRNA, 791nt long, targeting three ergosterol biosynthesis genes CYP51A, CYP51B, and CYP51C of *F. graminearum*, effectively inhibited the fungal growth both in local areas, where the dsRNA was sprayed and in non-sprayed distal leaf parts [17]. These results demonstrate that dsRNA can translocate within the plant. Topical application of dsRNA and sRNAs targeting *Dicer-like* (*DCL*) genes of *B. cinerea* (*BcDCL1* and *BcDCL2*) on the surface of tomato, strawberry, fox grape (*Vitis labrusca*), iceberg lettuce, onion, rose, and *Arabidopsis* leaves, effectively suppressed gray mold disease [11]. On the other hand, the capacity of exogenously applied dsRNAs to prevent and counteract infection of *B. cinerea* was tested on grapevine (*Vitis vinifera*). Three separate approaches for dsRNA delivery into plants were applied, namely, high-pressure spraying of leaves, petiole adsorption of dsRNAs, and postharvest spraying of bunches. The results demonstrated that, independently from the method of application, the exogenous method can decrease the virulence of *Botrytis cinerea* [18]. These successful experiments of exogenous application indicated that exogenously supplied dsRNA could form the basis for the development of a new tool aimed at protecting crops against fungal diseases.

The exogenous application of dsRNA can be very interesting also on horticultural produces at the postharvest stage [11] and against fungal pathogens, which are capable of producing mycotoxins very harmful to animal and human health [44,160]. Their control at the disposition stage is strictly limited to a few active ingredients due to residue concerns. With regard to postharvest pathogens, the halted growth of *B. cinerea* on the surface of fruits, vegetables, and flowers due to dsRNAs and sRNAs of *BcDCL1/2* [11] shows the potential of externally applied small RNA as a new generation of sustainable and environmentally friendly products for controlling postharvest pathogens. In addition, it should be recalled that post-harvest products are not exposed to open field environmental conditions such as UV light that promote degradation of dsRNAs and this makes them more suitable for protection during post-harvest.

## 5. Challenges of dsRNA-Based Products for Disease Management Strategy in Plants

Exogenous application of dsRNA molecules has been largely successful to induce RNAi (Table 3), and the studies outlined above highlight several critical aspects that need to be addressed before the development of RNAi-based products against fungal pathogens. Some considerations are required concerning the future application of exogenous RNA molecules against fungi and addressing the major issues that presently limit the viability of RNAi for fungal pathogen control.

### 5.1. Epigenetic Effect

As mentioned above, exogenous RNAi is an efficient transgene-free approach in modern crop protection platforms. In SIGS approaches, RNA molecules are externally applied on plants in order to selectively trigger the degradation of target mRNAs. However, once present in the plant cell, the applied dsRNAs may be processed by DCL4 into 21-nt siRNAs, which slice complementary mRNAs in a process termed post-transcriptional gene silencing [161], and by DCL2 into 22-nt siRNAs, which either recruit RNA-directed RNA polymerase 6 (RDR6) on the complementary mRNA for the generation of secondary siRNAs or repress mRNA’s translation [162,163]. Finally, DCL3 processes the dsRNA into 24-nt siRNAs, that are involved in RNA-directed DNA methylation (RdDM) of cognate DNA sequences [164]. Thus, in exogenous RNAi methods, the applied dsRNAs can trigger unexpected epigenetic alterations and lead to epigenetically modified plants. DNA methylation refers to the addition of a methyl group to the fifth carbon of the six-ring cytosine residue. DNA methylation was expected to be caused by DNA:DNA interactions for a long time, until a groundbreaking study showed that RNA:DNA interactions cause DNA methylation in viroid-infected tobacco plants, which was thus called RdDM [165]. Dubrovina and colleagues [147] applied in-vitro transcribed dsRNA targeting *GFP* and *NPTII* genes in transgenic *Arabidopsis thaliana* carrying a *GFP/NPTII* cassette. They observed that not only were *GFP* and *NPTII* mRNAs downregulated, but also DNA methylation occurred in the corresponding coding region 7 days after administration [136]. Therefore, the information from Dubrovina and colleagues [136] seem to reflect a more general mechanism and support a more careful consideration of possible epigenetic changes in the application of exogenous RNAi, because plants treated with exogenous dsRNAs may still contain no transgenes, but they are still epigenetically modified. In general, the occurrence of epigenetic changes in the genome after the application of exogenous RNAi should be resolved and clarified. This will help better interpret the exogenous RNAi data obtained.

### 5.2. Biosafety Considerations

Because of its sequence-dependent mode of action, there is increasing interest to use RNAi, both in academia and the commercial sector, in the management strategies for a large number of agricultural pests and pathogens as either in planta stable expression or in topical application [166]. RNAi-based plants have been already approved at the commercial level (corn and potato) and others are ready for submission (plum). The main issues for developing the risk assessment on these plants have been already defined [167]. The same biosafety approaches can be used to assess and approve new RNAi-based products for topical application. Below, we try to synthesize the most important aspects that need to be addressed in the risk assessment of plants during exogenous RNAi application. Although the binding of dsRNA/siRNA is believed to be highly specific [168], the siRNAs can bind to off-target genes that have sufficient sequence homology to the target gene [169]. The binding of siRNA somewhere else within the target genome may not be a problem, but concerns increase if off-target binding happens in non-target organisms.

However, to reduce possible effects on non-target species, it is possible to use the sequence-dependent nature of RNAi as an advantage to tailor the design of dsRNA sequences [147]. In fact, at the beginning of the development phase of the exogenous-RNAi mechanism, a thoughtful design of dsRNA will restrict the possibility of non-target effects due to sequence similarity. Designing a unique siRNA/dsRNA, which does not share high DNA identity with other genetic loci greatly limits the probability of off-target effects [170,171]. Current siRNA and dsRNA design guidelines for RNAi experiments suggest BLAST similarity searches (http://www.ncbi.nlm.nih.gov/BLAST (accessed on 4 March 2021)) [172] against sequence databases to pinpoint potential off-target genes to increase the probability that only the intended gene is targeted [173]. However, the BLAST algorithm was not specifically designed to assess RNAi off-target effects. Therefore, dedicated bioinformatics programs, like the open-access siRNA finder (si-FI) software (https://github.com/snowformatics/siFi21 (accessed on 4 March 2021); Lücketal, 2019), ERNAi (https://www.dkfz.de/signaling/e-rnai3/ (accessed on 4 March 2021)) and dsCheck (http://dscheck.rnai.jp/ (accessed on 4 March 2021)), can also be used to screen the candidate dsRNA/siRNA sequences for complementarity with other genes.

## 6. Future Prospects and Concluding Remarks

Food security is threatened by production constraints including diseases. Crop protection against pathogens relies mostly on the widespread use of chemical pesticides that are applied to the environment in large amounts yearly. Some of these chemicals have been in use for almost half a century. Therefore, there is a need for novel tools that are more sustainable and less detrimental to the environment. RNAi is a novel and promising method that is gaining pace as a technique to cope with pathogens in many economically important crop plants. Despite few limitations, the applicability of RNAi to improve crop resistance, especially against pathogens, is expected to be the most reliable and significant approach in the future, as shown by a plethora of studies. Generally, RNAi has emerged as one of the most promising potential control mechanisms for plant pathogens and insects. Although still a lot remains to be explored and understood about the molecular process of RNAi in plants and their pathogens, the current knowledge available and the studies reviewed in this paper have proved that exogenous RNAi technology is an essential tool for identifying gene functions and targeting critical genes to control plant pathogenic fungal development. In-planta stable expression offers a possible long-term stable resistance to diseases. In-planta stable expression offers the benefits of a long-term stable resistance to diseases, but it is clearly classified as a GMO and needs to follow rules applied for this type of modified plants [167]. Topical application, on the other hand, offers a more flexible solution for developing new dsRNA-based products to be used to protect crops in agricultural systems. Although information on external RNA uptake in fungi is limited, interesting progress has been achieved in *B. cinerea, F. asiaticum, F. graminearum, F. Oxysporum, M. phaseolina, M. fijiensis*, and *S. Sclerotiorum*. RNAi technology using the topical application of RNA molecules has emerged as a potential tool for improving various agronomically important plants. RNA-based biocontrol compounds are already under development and there is the perspective that new RNAi based formulates soon will reach the market, with a good cost-benefit balance for their application in different agriculture sectors. This objective now seems quite achievable considering the availability of first documents, the most important one from OECD (http://www.oecd.org/officialdocuments/publicdisplaydocumentpdf/?cote=env/jm/mono(2020)26&doclanguage=en (accessed on 4 March 2021)), which indicate risk assessment and regulatory approaches for these new RNAi-based products in line with those applied for the authorization of new biological pesticides [112].

To develop dsRNA-based products, besides the identification of effective dsRNA sequences, we need to develop appropriate formulates and delivering systems depending on the type of fungi and plants. Technological advancement in the field of biotechnology has offered new understandings to detect distinctive target genes. In fungi, the formulation, uptake, and processing of dsRNAs remains relatively undescribed. Analyzing the stability and delivery methods of dsRNAs, and more specifically the uptake of these dsRNAs into the target organism, remains ready for investigation. The delivery of dsRNA via nanoparticle complexes has novel potential for crop protection against pests, especially those refractories to RNAi. The topical use of dsRNA/nanoparticle complexes is expected to be the future of RNAi-mediated control of pests/pathogens without genetic modification of crops. Although carrier compounds considerably facilitate RNA delivery, they are also quite expensive and/or difficult to synthesize. Biosafety approaches already adopted to approve RNAi-based plants can be used for developing the risk assessment for new dsRNA-based products. Existing legislation should be implemented to consider the approval of new dsRNA-based products. Taking into account these aspects, we can think of a very important role in the development of this technology to improve the systems of protection of plants from diseases in a more compatible way with the environment, as foreseen by the new lines expected from the green deal indicated by Europe and of interest in the world [166].

## Figures and Tables

**Table 1 plants-10-00650-t001:** Representative potential target genes tested for controlling pathogenic fungi and oomycetes.

Species	Target Gene(s)	Host Plant	References
*Alternaria alternata*	*Putative hydrolase (ACTT2), a host-selective ACT-toxin*	Tangerine	[42]
	*Enoyl-reductase (ACTTS2), a host-selective ACT-toxin*	Tangerine	[43]
*A. flavus and A. parasiticus*	*Transcription factor (aflR)*	Corn and wheat	[44]
*A. flavus*	*aflS, aflR, aflC, pes1, aflep*	Peanut	[45]
	*aflR*	Maize	[46]
*Blumeria graminis f. sp. tritici*	*MLO*	Wheat	[47]
*Bipolaris oryzae*	*Polyketide synthase gene (PKS1)*	-	[48]
*Blumeria graminis*	*Avira10*	Barley and wheat	[40]
	*BEC1011, BEC1054, BEC1038,* *BEC1016, BEC1005, BEC1019,* *BEC1040, and BEC1018*	Barley	[49]
*Botrytis cinerea*	*Superoxide dismutase (BCSOD1)*	French bean	[50]
	*Dicer-like 1* and *Dicer-like 2*	Arabidopsis, tomato, strawberry, grapes, lettuce, onion, and rose	[11]
*Bremia lactucae*	*Cellulose synthase 1, Highly abundant message #34 (HAM34)*	lettuce	[24]
*Cladosporium fulvum*	*Hydrophobin gene (HCf-1)*	-	[51]
	*First exons of six hydrophobin coding genes*	-	[52]
*Cochliobolus sativus*	*GFP, a host-selective toxin (ToxA) and a polyketide synthase (CsPKS1)*	Wheat	[41]
*Colletotrichum gloeosporioides*	*Transcription factor (PAC1)*	-	[53]
*Fusarium culmorum*	*FcGls1*	Wheat	[54]
*Fusarium graminearum*	*Transcription factor (Tri6)*	Corn and wheat	[44]
	*Cytochrome P450 lanosterol C-14α-demethylase genes CYP51A, CYP51B and CYP51C*	Arabidopsisand barley	[17]
	*Chs3b*	Wheat	[55]
*Fusarium oxysporum f. sp.* *cubense (fusarium wilt)*	*Velvet, Fusarium transcription factor 1*	Banana	[56]
*F. oxysporum f. sp.*	*FRP1, FOW2, OPR*	Arabidopsis	[57]
*Fusarium solani f.sp. pisi*	*ß (1,3)-D-glucan synthase (FsFKS1)*	-	[58]
*Fusarium solani*	*Chitosanase (CSN1)*	Pea	[59]
*F. verticillioides*	*GUS (ß glucuronidase)*	Tobacco	[60]
*Glomus species*	*Monosaccharide transporter 2*	Potato	[61]
*Magnaporthe oryzae*	*MPG1 and PKS-like gene*		[62]
	*37 genes involved in calcium signalling*	Barley and wheat	[63]
*Melampsora lini*	*Effector protein (AvrL567)*	Flax	[64]
*Moniliophthora perniciosa*	*GFP, hydrophobin (MpHYD3) and 1-cys peroxiredoxin (MpPRX1)*	-	[65]
*Mucor circinelloides*	*Carotenogenic gene (carB)*	-	[66]
*Mycosphaerella fijiensis, Fusarium oxysporum*	*Nuclear condensin, coatomer alpha,* *DNA-directed RNA polymerase, actin cortical patch 2/3, coatomer zeta, CAP* *Methyltransferase, GTP ASE binding protein, proteasome PRE4, Ribosomal RNA, DNA Polymerase alpha/delta subunit, Adenylase cyclase, Protein kinase C, FRQ-interacting RNA helicase*	-	[67]
*Ophiostoma novo-ulmi*	*Endopolygalacturonase (Epg1)*	-	[68]
*Puccinia triticina*	*MAPK, cyclophilin (CYC1),* *and a calcineurin (CNB)* *regulatory subunit gene*	Wheat	[69]
*Puccinia striiformis f. sp. tritici*	*PsCPK1, PsFuz7*	Wheat	[36]
*Phytophthora infestans*	*G-protein b-subunit encoding gene (Pigpb1)*	Potato	[70]
	*Cdc 14 coding gene (PiCdc14)*	-	[71]
	*G-protein a-subunit gene (Pigpa1)*	Potato	[72]
	*cdc14*	-	[73]
*Phytophthora infestans*	*bZIP transcription factor (Pibzp1)*	Tomato	[74]
	*Nuclear LIM interactor-interacting factors (NIFC1 andNIFC2)*	Tomato	[75]
	*Inf1*		[76]
	*Putative glycosylated protein (Pihmp1)*	Potato	[77]
	*Putative ATP-dependent DEAD-box RNA-helicase gene (Pi-RNH1)*	Potato	[78]
	*Four members of the CesA encoding for cellulose synthase genes*	Potato	[79]
	*Effector protein (PiAVR3a)*	Tobacco and potato	[80]
	*SYR1*	Potato	[81]
	*Cutinase*	Potato	[82]
	*Dicer-like (Pidcl1), Argonaute (Piago1/2), Histone deacetylase (Pihda1)*	Potato	[83]
	*G protein* β*-subunit* (*GPB1*), *Cellulose synthase A2, Pectinesterase, Glyceraldehyde 3-phosphate*	Potato	[16]
	*DCL1, HMP1-, PGB1-,* and *DCTN1+SAC1*	Potato	[84]
*P. parasitica var.nicotianae*	*A coding gene considered to be involved in cellulose-binding (CB), elicitor (E) of defence in plants and lectin (L)-like activities (CBEL)*	Tobacco	[85]
	*GST*	Tobacco	[86]
*Phytophthora nicotianae,* *Peronospora tabacina*	*Cutinase*	Tobacco	[82]
*Phytophthora sojae*	*Heterotrimeric G-protein a subunit (PsGPA1)*	Soybean	[87]
	*C2H2 zinc finger transcription factor (PsCZF1)*	Soybean	[88]
	*MAP kinase encoding gene*	Soybean	[89]
	*(PsSAK1)*	Soybean	[90]
	*Putative seven-transmembrane G-protein-coupled receptor (GPR11)*	Soybean	[91]
	*PsYKT6, a conserved member gene of the soluble N-ethylmaleimide-sensitive factor attachment protein receptors (SNAREs)*	Tobacco and soybean	[92]
	*Crinkling- and necrosis-inducing proteins (CRN) (PsCRN63 and PsCRN115)*	Glycine max	[93]
*Puccinia striiformis f. sp.tritici*	*PsCPK1, PsFuz7*	Wheat	[36]
*Puccinia striiformis f. sp.tritici*	*PsCNA1 and PsCNB1*	Barley and wheat	[11]
*Puccinia triticina*	*MAP kinase (PtMAPK1), cyclophilin (PtCYC1), calcineurin B (PtCNB)*	Wheat	[69]
*Sclerotinia sclerotiorum*	*B regulatory subunit (rgb1) of 2A* *phosphoprotein phosphatase (PP2A)*	Tomato	[93]
	*Chitin synthase*	Tobacco	[94]
*Ustilago hordei*	GUS and mating-type gene (*bW*)		[95]
*Verticillium dahliae*	*Ave1, SIX gene expression 1 (Sge1) and necrosis and ethylene-inducing-like protein (NLP1)*	Tomato and Arabidopsis	[96]
	*V. dahliae hygrophobins1*	Cotton	[97]
*Verticillium longisporum*	*Chorismate synthase (Vlaro2)*	Arabidopsis and rapeseed	[98]
*Venturia inaequalis*	*Trihydroxynaphthalene reductase (THN)*	Apple	[39]

**Table 2 plants-10-00650-t002:** Advantages and disadvantages of different methods of double-stranded RNAs (dsRNAs)/small interfering RNAs (siRNAs) production.

Methods	Advantage	Disadvantages	Fungal Pathogen Tested with the Technology and References
**In Vitro**			
Enzymatic synthesis	Less expensive No need to test individual siRNA separately	Purity and specificity are variable	[11,34,38]
Chemical synthesis	Fast/Rapid High purity	Expensive	
**In vivo**			
*Escherichia coli/ Pseudomonas syringae*	Produce large quantities of dsRNAs at low cost	Labor intensive	[18]
*Yarrowia lipolytica*	Produce large quantities of dsRNAs at low cost	Labor intensive	

**Table 3 plants-10-00650-t003:** Summary of exogenously applied RNA molecules to plant pathogenic fungi/ascomycetes.

Host Plant	Species	Target Gene(s)	Role(s) of Target(s) Gene(s)	Method of Production	References
**Cereals**					
Barley	*Fusarium graminearum*	*CYP51A, CYP51B,* and *CYP51C*	Ergosterol biosynthesis	*In vitro* (MEGA script® RNAi Kit	[17]
Barley	*Fusarium asiaticum*	*ß2 tubulin*	Fungal growth	*In vitro* (MEGA script® RNAi Kit)	[37]
Barley	*Fusarium graminearum*	*ARGONAUTE* and *DICER*	Fungal vegetative and generative growth, mycotoxin production, antiviral response	*In vitro* (MEGA script® RNAi Kit)	[38]
Rice	*Rhizoctonia solani*	*DCTN1, SAC1, polygalacturonase (PG)*	Vesicle trafficking pathway genes and virulence factor	*In vitro (MEGA script® RNAi Kit)*	[84]
Wheat	*Fusarium asiaticum*	*Myosin 5 gene*	Cytokinesis and actin filaments organization	*In vitro* (MEGA script® RNAi Kit)	[106]
Wheat	*Fusarium asiaticum*	*ß2 tubulin*	Fungal growth	*In vitro* (MEGA script® RNAi Kit)	[37]
Wheat	*Fusarium graminearum*	*RdRP1, AGO1, QDE3, QIP, AGO2,* *DCL1, RdRP2, RdRP3, RdRP4,* and *DCL2*	Sexual reproduction AGO generative development DCL1		[107]
**Vegetable**					
Cucumber	*Fusarium asiaticum*	*ß2 tubulin*	Fungal growth	*In vitro* (MEGA script® RNAi Kit)	[37]
Tomato	*Aspergillus niger*	*VPS51, DCTN1, SAC1, pgxB,*	Vesicle trafficking pathway genes and virulence factor	*In vitro (MEGA script® RNAi Kit)*	[84]
	*Botrytis cinerea*	*DCL1* and *DCL2*	Effectors	*In vitro* (MEGA script® RNAi Kit)	[11]
		*VPS51, DCTN1, SAC1*	Vesicle trafficking pathway genes	*In vitro (MEGA script® RNAi Kit)*	[84]
	*Colletotrichum* *gloeosporioides*	*DCL 1-2, VPS51, DCTN1, SAC1*	Effectors and vesicle trafficking pathway genes	*In vitro (MEGA script® RNAi Kit)*	[84]
Lettuce	*Botrytis cinerea*	*DCL1* and *DCL2*	Effectors	*In vitro* (MEGA script® RNAi Kit)	[11]
		*VPS51, DCTN1, SAC1*	Vesicle trafficking pathway genes	*In vitro (MEGA script® RNAi Kit)*	[84]
	*Sclerotinia sclerotiorum*	*VPS, DCTN1, SAC1*	Vesicle trafficking pathway genes	*In vitro (MEGA script® RNAi Kit)*	[84]
Collard green	*Sclerotinia sclerotiorum*	*VPS, DCTN1, SAC1*	Vesicle trafficking pathway genes	*In vitro (MEGA script® RNAi Kit)*	[84]
Onion	*Botrytis cinerea*	*DCL1* and *DCL2*	Effectors	*In vitro* (MEGA script® RNAi Kit)	[11]
**Oil Crops**					
Soya	*Fusarium asiaticum*	*ß2 tubulin*	Fungal growth	*In vitro* (MEGA script® RNAi Kit)	[37]
Canola	*Sclerotinia sclerotiorum*	59 target genes	Cell wall modification, mitochondria, ROS response, protein modification, pathogenicity factors, transcription, splicing, and translation	*In vitro* (MEGA script® RNAi Kit)	[34]
**Fruit Crops**					
Apple	*Aspergillus niger*	*VPS51, DCTN1, SAC1, pgxB,*	Vesicle trafficking pathway genes and virulence factor	*In vitro (MEGA script® RNAi Kit)*	[84]
	*Colletotrichum gloeosporioides*	*DCL 1-2, VPS51, DCTN1, SAC1*	Effectors and vesicle trafficking pathway genes	*In vitro (MEGA script® RNAi Kit)*	[84]
Banana	*Mycosphaerella fijiensis,* *Fusarium oxysporum*	*Nuclear condensing, Coatomer alpha, DNA-directed RNA polymerase, ARP 2/3, Coatomer zeta, Cap methyltransferase, GTPase-binding protein, Proteasome Pre4, Ribosomal RNA, DNA polymerase alpha subunit, DNA polymerase delta* *Subunit, Adenylase cyclase, Protein kinase C, FRQ-interacting RNA* *helicase*	Spore germination	*In vitro* (MEGA script® RNAi Kit)	[67]
Cherry	*Colletotrichum* *gloeosporioides*	*DCL 1-2, VPS51, DCTN1, SAC1*	Effectors and vesicle trafficking pathway genes	*In vitro (MEGA script® RNAi Kit)*	[84]
Grape	*Botrytis cinerea*	*BcCYP51, Bcchs1,* and *BcEF2*	Elongation factor, ergosterol and chitinase biosynthesis	*In vivo* (HT115 (DE3) *E. Coli)*	[18]
		*DCL1* and *DCL2*	Effectors	*In vitro* (MEGA script® RNAi Kit)	[11]
		*VPS51, DCTN1, SAC1*	Vesicle trafficking pathway genes	*In vitro (MEGA script® RNAi Kit)*	[84]
	*Aspergillus niger*	*VPS51, DCTN1, SAC1, pgxB,*	Vesicle trafficking pathway genes and virulence factor	*In vitro (MEGA script® RNAi Kit)*	[84]
Strawberry	*Botrytis cinerea*	*DCL1* and *DCL2*	Effectors	*In vitro* (MEGA script® RNAi Kit)	[11]
**Flowers**					
Rose	*Botrytis cinerea*	*DCL1* and *DCL2* *DCL1* and *DCL2*	Effectors	*In vitro* (MEGA script® RNAi Kit)	[11]
		*VPS51, DCTN1, SAC1*	Vesicle trafficking pathway genes	*In vitro (MEGA script® RNAi Kit)*	[84]
**Model Plant**					
Arabidopsis	*Botrytis cinerea*	*DCL1* and *DCL2*	Effectors	*In vitro* (MEGA script® RNAi Kit)	[11]
Arabidopsis	*Sclerotinia sclerotiorum*	59 target genes	Differentially upregulated genes	*In vitro* (MEGA script® RNAi Kit)	[34]
Arabidopsis	*Fusarium graminearum*	*CYP51*	Ergosterol biosynthesis	*In vitro* (MEGA script® RNAi Kit)	[108]
Arabidopsis	*Verticillium dahliae*	*DCL 1-2, DCTN1, SAC1*	Effectors and vesicle trafficking pathway genes	*In vitro (MEGA script® RNAi Kit)*	[84]
Arabidopsis	*Macrophomina phaseolina*	*C* *hitin synthase (MpCHS) gene*	Catalyze the β-1,4 polymerization of *N-acetylglucosamine*		[109]

## Data Availability

The data presented in this study are available in the article.

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
