# Peer review of "RNA Interference Strategies for Future Management of Plant Pathogenic Fungi: Prospects and Challenges"

_plants, 2021, doi:10.3390/plants10040650_

Round 1

Reviewer 1 Report

Abstract just inform what is the content of the manuscript. It is missing crucial ideas or conclusions based on the review. It needs some information that can be cited if somebody just has only the abstract.

The manuscript includes a lot of very important information. All chapters are perfect compilation (so also the language is automatically good because the sentences were based on these data), and very informative.

I recommend the authors to pay more attention to the following:

  • Pathogenic fungi names were almost not mentioned in chapter 2 even all names are mentioned in Tables. I would prefer some more particular examples directly in chapter 2.
  • Chapter Conclusions: It is just a recommendation, but probably it will be interesting to mention in conclusions some fungal species, that could be practically controlled by RNA technologies sooner than the other. In which fungus it was achieved the highest progress until now? Or which taxonomic pathogen group?
  • Authors are experts or genetics, but my feeling is that team needs a plant pathologist who can ask the questions like “which pathogen, when … could be used”. Maybe it would be interesting to speak about a particular pathogen and then to discuss if there is any progress to control it with RNA technologies.

The name of the manuscript is “RNA Interference strategies for future management of plant pathogenic fungi: Prospects and Challenges”. It is necessary to emphasize the words “plant pathogenic fungi”. The nice tables are good information but they need more discussion.

Author Response

 Below the answers to reviews comments:

  1. Abstract just inform what is the content of the manuscript. It is missing crucial ideas or conclusions based on the review. It needs some information that can be cited if somebody just has only the abstract.

Thank you for your comment, we have improved the abstract based on your valuable comments. The following paragraphs are incorporated in the abstract as follows:

"…..Recent studies showed that the exogenous application of double-stranded RNA (dsRNA) molecules on plants targeting fungal growth and virulence related genes provided disease attenuation of pathogens like Botrytis cinerea, Sclerotinia sclerotiorum and Fusarium graminearum in different hosts. Such results highlight that the exogenous RNAi holds great potential for RNAi-mediated plant pathogenic fungal disease control. Production of dsRNA can be possible by using either in vitro or in vivo synthesis.. "

"……Potential challenges that are faced while developing a RNAi strategy for fungal pathogens, such as off-target and epigenetic effects, with their possible solutions are also discussed."

  1. Pathogenic fungi names were almost not mentioned in chapter 2 even all names are mentioned in Tables. I would prefer some more particular examples directly in chapter 2.

Thank you for your comment, some pathogenic fungi names are mentioned in chapter 2 starting from line 29 as follows :

"…..different mold fungi, such as Botrytis cinerea, Neurospora crassa and Sclerotinia sclerotiorum [11,18,33,34], blast, blight and rust fungi, such as Fusarium asiaticum, Fusarium graminearum, Magnaporthe oryzae and Puccinia striiformis f. sp. tritici [17,35,36,37,38], mildew and others, such as Blumeria graminis, Cochliobolus sativus and Venturia inaequalis…."

  1. Chapter Conclusions: It is just a recommendation, but probably it will be interesting to mention in conclusions some fungal species, that could be practically controlled by RNA technologies sooner than the other. In which fungus it was achieved the highest progress until now? Or which taxonomic pathogen group?

Thank you for your comment and we have incorporated your comment in the conclusion starting from line 20 … as follows :

"……Although information on external RNA uptake in fungi is limited,  interesting progress has been achived  in B. cinerea, F. asiaticum, F. graminearum, F. oxysporum, M. phaseolina, M. fijiensis and S. Sclerotiorum…".

  1. Authors are experts or genetics, but my feeling is that team needs a plant pathologist who can ask the questions like “which pathogen, when … could be used”. Maybe it would be interesting to speak about a particular pathogen and then to discuss if there is any progress to control it with RNA technologies.

Thank you again for your comments the authors are both genetics and plant pathologist, in chapter 4 we address the issues raised in your comments.

  1. The name of the manuscript is “RNA Interference strategies for future management of plant pathogenic fungi: Prospects and Challenges”. It is necessary to emphasize the words “plant pathogenic fungi”. The nice tables are good information but they need more discussion.

We thank the reviewer for this comment If we address your question perfectly, the tables are inferred in the text

Reviewer 2 Report

 Dear authors,

I read with interest your article about the future of plant protection.
As a forest pathologist, I enviously read about agricultural and horticultural applications.
However, has no one in the world undertaken research to protect plants of forest-forming species?
Especially, that some of the mentioned pathogens, e.g.
such as Botrytis cinnerea, are met in forest nurseries, too .
If so, it is worth mentioning in the article because it is the natural forest ecosystems that should be first of all free from the use of chemicals in forest protection in the future.
I wonder how large concerns producing plant protection means (controling chemical and biological products) will react when created environmentally safe alternatives will be implemented in large scale? (if feasible). Will these types of RNAi-based preparations require registration like other plant protection products to date? How to assess their potential impact on the environment, including humans and animals?
Will we be able to feed the growing human population when withdrawing chemicals
from primary production?
If the authors think that it is worth mentioning these dilemmas in the discussion, especially in the aspect of sustainable agriculture, horticulture and forestry, then please.
If not, I still think that the topic raised in the article is very important and worth discussing on an international forum. Congrats.

Author Response

Below the answer to the referee's comments:

  1. As a forest pathologist, I enviously read about agricultural and horticultural applications.

Thank you for your comment. We are sorry if we have not extended the review to forest pathology topics but we partly lack the expertise in this area and in reality there are still not many studies available on specific applications of dsRNA for the control of pathogenic fungi in forest plants. We can only try to give some answers to your comments, but we do not believe it is useful to include them in the review.

  1. However, has no one in the world undertaken research to protect plants of forest-forming species? 3.Especially, that some of the mentioned pathogens, e.g. such as Botrytis cinnerea, are met in forest nurseries, too.4. If so, it is worth mentioning in the article because it is the natural forest ecosystems that should be first of all free from the use of chemicals in forest protection in the future.

The only studies were able to find summarizing applications on of RNAi in forest tree is included in this chapter prepared by Matthias Fladung, Hely Haggman and S. Sutela:  Application of RNAi technology in Forest Trees, on the book under publication by CABI - https://www.cabi.org/bookshop/book/9781789248890/?utm_source=Books&utm_medium=ext_website&utm_campaign=1lsb_author_mezzetti_PS_12&utm_term=Mar20&utm_content=Org. But in this chapter are reported only preliminary results on in planta RNAi silencing of reporter and flowering genes. We do not think that the addition of this quote will help improve this more specific review on the direct application of dsRNA for fungal control.

  1. I wonder how large concerns producing plant protection means (controlling chemical and biological products) will react when created environmentally safe alternatives will be implemented in large scale? (if feasible). Will these types of RNAi-based preparations require registration like other plant protection products to date? How to assess their potential impact on the environment, including humans and animals?

We thank the reviewer for these comments, and we tried to address the issues raised in the conclusion by implementing the manuscript with the following text  starting from line 16 and line 23  :

In-planta stable expression offers the benefits of a long-term stable resistance to diseases but it is clearly classified as a GMO and need to follow rules applied for this type of modified plants [110].

 RNA-based biocontrol compounds are already under development and there is the perspective that new RNAi based formulates soon will reach the market, with a good cost-benefit balance for their application in different agriculture sectors. This objective now seems quite achievable considering the availability of first documents, which indicate risk assessment and regulatory approaches for these new RNAi-based products in line with those applied for the authorization of new biological pesticides [51].

6.Will we be able to feed the growing human population when withdrawing chemicals from primary production?

  1. If the authors think that it is worth mentioning these dilemmas in the discussion, especially in the aspect of sustainable agriculture, horticulture and forestry, then please.

We thank the reviewer for these comments, we agree that feeding the growing population is a great dilemma for the future and for sure new solutions have to be found and supported by appropriate policies for introducing more sustainable and safe products. Although our review is not so ambitious to offer definitive options to solve this problem, we think we can contribute by highlighting the importance of better understanding the potential of RNAi technology in improving the sustainability of agricultural systems. To underline this topic at the end of the manuscript we only added the following reference : Taning, C.N.T., Mezzetti, B., Kleter, G., Smagghe, G., Baraldi, E., 2020. Does RNAi-Based Technology Fit within EU Sustainability Goals? Trends in Biotechnology, 12, 10.1016/j.tibtech.2020.11.008.

  1. If not, I still think that the topic raised in the article is very important and worth discussing on an international forum. Congrats.

We thank the reviewer for these really positive comments, encouraging a lot our group to continue with this study

Reviewer 3 Report

1.A recent biorxiv study “Spray-induced gene silencing for disease control is dependent on the efficiency of pathogen RNA uptake” has determined the efficiency of RNA uptake in multiple pathogenic and non-pathogenic fungi, and an oomycete pathogen. They found that Botrytis cinerea, Sclerotinia sclerotiorum, Rhizoctonia solani, Aspergillus niger, and Verticillium dahlia can take up RNAs with high efficiency, but no uptake in Colletotrichum gloeosporioides, and weak uptake in a beneficial fungus, Trichoderma virens. For the oomycete plant pathogen, Phytophthora infestans, RNA uptake was limited, and varied across different cell types and developmental stages. The authors can add this information in the section 4 when talking about the dsRNA uptake by fungi.

  1. The abbreviation in the manuscript is a little messy. For example, “RNA interference” and “RNAi” have been mixed in the whole manuscript, it will be better to use the same one to make the whole manuscript easy to read. Another one is “RNA-directed DNA methylation (RdDM)”, which is repeated twice in section 5.1. I think maybe it is because multiple authors wrote this manuscript, so please double check the whole manuscript again.
  2. In table 1, many host plant species are missing, please add them.

Author Response

Below the answers to the referee's  comments: 

1.A recent biorxiv study “Spray-induced gene silencing for disease control is dependent on the efficiency of pathogen RNA uptake” has determined the efficiency of RNA uptake in multiple pathogenic and non-pathogenic fungi, and an oomycete pathogen. They found that Botrytis cinerea, Sclerotinia sclerotiorum, Rhizoctonia solani, Aspergillus niger, and Verticillium dahlia can take up RNAs with high efficiency, but no uptake in Colletotrichum gloeosporioides, and weak uptake in a beneficial fungus, Trichoderma virens. For the oomycete plant pathogen, Phytophthora infestans, RNA uptake was limited, and varied across different cell types and developmental stages. The authors can add this information in the section 4 when talking about the dsRNA uptake by fungi.

Thank you for your valuable comments and we added your comment in section 4. Starting from line 32 as follows :

"……. One of the few recent studies reported that various beneficial or pathogenic fungal and oomycetes organisms have diverse capacity to adsorb fluorescein-labeled dsRNA from the environment, and this competence seems to have an influence on the efficacy of the RNAi when virulence-related gene were targeted through SIGS approach for the defense of the hosts. The authors showed that Colletotrichum gloeosporioides cannot uptake dsRNA, whereas in Trichoderma virens and Phytophthora infestans RNA uptake was limited. The situation is different in Botrytis cinerea, Sclerotinia sclerotiorum, Rhizoctonia solani, Aspergillus niger, and Verticillium dahliae in which fluorescent dsRNAs are already inside the fungal cells within 6 hours after administration of specific long dsRNA [75]….."

  1. The abbreviation in the manuscript is a little messy. For example, “RNA interference” and “RNAi” have been mixed in the whole manuscript, it will be better to use the same one to make the whole manuscript easy to read. Another one is “RNA-directed DNA methylation (RdDM)”, which is repeated twice in section 5.1. I think maybe it is because multiple authors wrote this manuscript, so please double check the whole manuscript again.

Thank you for the comments and suggestion and we fixed the disorder concerning "RNA interference" and "RNA-directed DNA methylation " throughout the manuscript.

3.In table 1, many host plant species are missing, please add them.

Thank you for the comments, we included host plant species for some but still some are missing because the researchers did not use plant species as host.